# Efficient Protein Optimization via Structure-aware Hamiltonian Dynamics

## Abstract

The ability to engineer optimized protein variants has transformative potential for biotechnology and medicine. Prior sequence-based optimization methods struggle with the high-dimensional complexities due to the epistasis effect and the disregard for structural constraints. To address this, we propose HADES, a Bayesian optimization method utilizing Hamiltonian dynamics to efficiently sample from a structure-aware approximated posterior. Leveraging momentum and uncertainty in the simulated physical movements, HADES enables rapid transition of proposals toward promising areas. A position discretization procedure is introduced to propose discrete protein sequences from such continuous state system. The posterior surrogate is powered by a two-stage encoder-decoder framework to determine the structure and function relationships between mutant neighbors, consequently learning a smoothed landscape to sample from. Extensive experiments demonstrate that our method outperforms state-of-the-art baselines in in-silico evaluations across most metrics. Remarkably, our approach offers a unique advantage by leveraging the mutual constraints between protein structure and sequence, facilitating the design of protein sequences with similar structures and optimized properties.

## 1 Introduction

Designing proteins with improved fitness is a fundamental but challenging task in protein engineering. The search space of protein variants is vastly expanded as the length of the protein sequence grows, which is typically $20^L$ for a sequence composed of $L$ amino acids for 20 types of amino acids. Optimizing protein fitness is difficult due to the complex epistasis effect, which creates a rugged, multi-peaked landscape (Poelwijk et al., 2011). Traditional directed evolution approaches rely on random variation combined with screening and selection, without incorporating models to understand the sequence-function relationship (Arnold, 1998). This process is costly and time-consuming, limiting the exploration to only a small fraction of possible mutations, even with the advent of reasonably high-throughput techniques.

The challenge of identifying novel protein designs for maximal fitness has driven scientists to adopt machine learning approaches, starting with (Fox et al., 2007) and followed by many others, increasingly utilizing in physics- and machine learning-based exploration beyond traditional experimental methods (Sinai et al., 2020; Frey et al., 2024; Gruver et al., 2024). Recent advancements in protein language models and structural-informed models also offer new opportunities for modeling protein mutational effects and generalizing fitness knowledge across different datasets(Zheng et al., 2023; Notin et al., 2023; Hie et al., 2024).

The primary research challenge in designing an iterative protein engineering strategy is navigating the vast combinatorial space to uncover the sequence-to-function landscape and find the optimal sequences (Romero & Arnold, 2009). This daunting task can be tackled by employing a black-box optimization approach, where the inputs are optimized to get better outputs with limited and often expensive black-box oracle calls. However, protein landscapes are notoriously non-smooth, with fitness levels that can drastically change due to a single mutation, and most mutants have poor fitness. As a result, existing machine learning methods often falter, struggling with noisy fitness landscapes that lead to false positives (Kirjner et al., 2023) and failing to venture beyond local optima due to inefficient exploration (Brookes et al., 2019).

Compared to sequence-based searching strategies, leveraging structural information offers a more natural approach by providing useful constraints for navigating the noisy fitness landscape, given the decisive role of 3D structure in determining protein function (Branden & Tooze, 2012). Therefore, when optimizing a protein for a desired function, designing it in structure latent space seems more direct, allowing the use of gradient-based sampling methods in conjunction with carefully engineered potentials (Trippe et al., 2022; Watson et al., 2023). However, a drawback of this approach is that the optimized structure must be converted back to an amino acid sequence for synthesis(Dauparas et al., 2022), while there is no guarantee that the optimized structure can be realized by an actual sequence. Additionally, current structural models are computationally intensive and constrained by the scarcity of high-quality structural data, posing challenges to optimization and search.

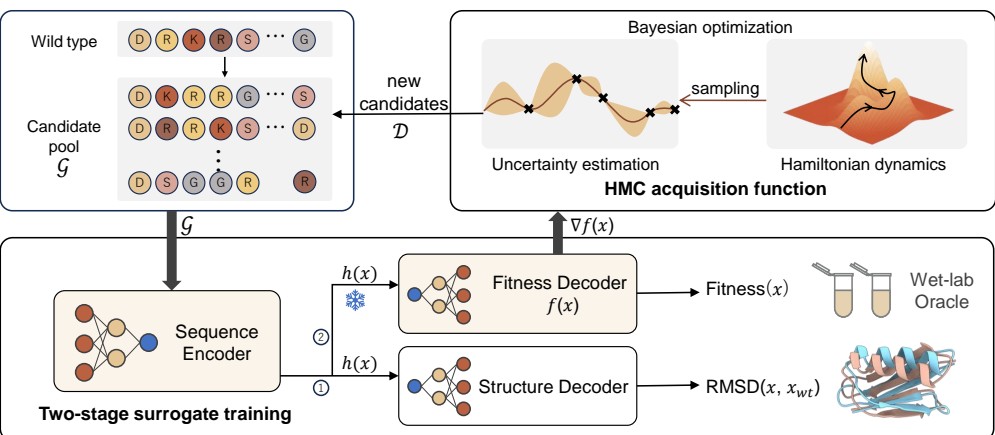

Figure 1: The overall architecture of the proposed HADES framework.

To address these issues, we propose a new learning-based protein engineering framework that combines Bayesian Optimization and **HA**miltonian dynamics for protein **D**irected **E**volution in a **S**tructure-informed manner (**HADES**, Figure 1). Our motivation is based on the mutual constraints between protein sequence and structure. The distribution of conformational perturbations among a series of mutants can serve as prior information to model a smoothed landscape of protein function, potentially enhancing the efficiency of gradient-based algorithms for exploration and sampling. We achieve this by designing a surrogate fitness predictor powered by a two-stage encoder-decoder to learn the structure perturbations distilled from ESMFold (Lin et al., 2023) as the prior. To address the inefficiency in exploring high-dimensional sequence space, we introduce an acquisition function based on Hamiltonian Monte Carlo that proposes distant samples for the Metropolis-Hastings algorithm in MCMC sampling steps by introducing momentum variable in a continuous state space. We further combine the sampling process under Bayesian optimization with estimation of model uncertainty to balance between exploration and exploitation. Our approach offers a significant advantage by leveraging the mutual constraints between protein structure and sequence, facilitating the design of protein sequences with similar structures and enhanced properties.

## 2 RELATED WORK

**Machine-learning-assisted directed evolution** Machine-learning accelerates the laboratory directed evolution by learning a sequence-to-function mapping and propose promising samples (Yang et al., 2019). Recent methods are based on paradigm of model-based black-box optimization, where different sequence-function models and exploration algorithms are proposed for better modeling and exploration of the function landscape based on sequence mutations (Hansen & Ostermeier, 2001; Brookes et al., 2019; Jain et al., 2022; Ren et al., 2022; Song & Li, 2023; Frey et al., 2024). Brookes & Listgarten (2018) and Brookes et al. (2019) adaptively sample sequences combining unsupervised generative models and black box predicted models. Ren et al. (2022) prioritizes the proximal exploration for high-fitness mutants with low mutation counts and design a mutation factorization network to predict low-order mutational effects. Kirjner et al. (2023) employs a graph-based smoothing to remove noisy gradients in the protein sequence landscape. Gruver et al. (2024) proposes a diffusion-

optimized sampling strategy for controllable categorical diffusion in discrete sequence space. Unlike previous methods, we perform a structure-informed search over continuous sequence space. This allows us to construct a smooth energy surrogate, ensuring strong performance guarantees.

**Hamiltonian mechanics and MCMC** Markov Chain Monte Carlo defines a transition kernel to explore target distribution, and Metropolis-Hastings algorithm is a common method to automatically construct appropriate transitions. For high-dimensional spaces, MCMC with Hamiltonian dynamics is an effective way to make large jumps away from the initial point (Neal, 2012b; Girolami & Calderhead, 2011). A key to its usefulness is that Hamiltonian dynamics preserves volume which can be exactly maintained even when the dynamics is approximated by discretizing time. A special case of implementation which takes a single leapfrog step in HMC is the Langevin Monte Carlo (LMC) algorithm, and the stochastic gradient version, stochastic gradient langevin dynamics, has been widely studied and applied (Welling & Teh, 2011; Chen et al., 2014; Frey et al., 2024). Compared to other MCMC works, our approach introduces an extra position discretization procedure to propose discrete protein sequence from a continuous state system.

## 3 BACKGROUND

In this study, we address the challenge of machine-learning-guided protein directed evolution within a black-box optimization framework. The process initiates with a single wild type protein sequence, denoted as $x_{wt}$, of length $L$. A protein sequence is represented as a concatenation of one-hot vectors, $x_{wt} = (x_1, x_2, ..., x_L)$, each vector corresponding to an amino acid. The ground-truth fitness of an unknown protein is assessed through wet-lab evaluations, functioning as the black-box oracle $\mathcal{F}(x)$.

The primary objective of this research is to engineer protein variants exhibiting enhanced fitness scores, particularly under the constraint of limited oracle evaluations. The directed evolution process is conducted iteratively: in each iteration $n$, where $n \in \{1, 2, ..., N\}$, a batch of protein variants $\mathcal{D}_n$, with $|\mathcal{D}_n| = K$, is analyzed by the black-box oracle to determine the true fitness scores $\mathcal{G}_n = \{(x, \mathcal{F}(x)), x \in \mathcal{D}_n\}$. The optimization sequence concludes after $N$ rounds of oracle calls.

Typically, a surrogate fitness predictor $f(x)$ is trained to approximate the black-box oracle, utilizing samples collected through an acquisition function. This function strategically proposes new, promising protein variants for subsequent rounds. Operating under a fixed budget of $K$ oracle queries per round, the acquisition process must effectively balance exploration of the expansive, high-dimensional sequence space with the exploitation of regions potentially close to optima. The surrogate fitness predictor, serving as an approximate posterior, provides fitness predictions that guide the acquisition function in selecting promising variants for further evaluation.

**Hamiltonian dynamics and MCMC** Markov Chain Monte Carlo (MCMC) with the Metropolis-Hastings algorithm is a widely-used technique for generating samples from an approximate posterior distribution. For an unknown target distribution $\pi(x)$, proposal distribution is define by $Q(x^{'}|x)$, and the probability to accept a proposal is $\min(1, \frac{Q(x|x^{'})\pi(x^{'})}{Q(x^{'}|x)\pi(x)})$.

However, in high-dimensional target distributions, the naive proposals in Metropolis-Hastings MCMC (MH-MCMC) often lead to low acceptance rates. Hamiltonian Monte Carlo (HMC), initially developed for molecular simulation and later adapted for use in statistics and neural network models, addresses this issue by leveraging Hamiltonian dynamics to propose new samples (Alder & Wainwright, 1959; Neal, 2012a).

A Hamiltonian system characterizes the energy of a frictionless particle as the sum of its potential energy $U(q)$ and kinetic energy $K(p)$:

$$H(q, p) = U(q) + K(p)$$

Here, $q$ represents the particle's position, and $p$ represents its momentum. As the particle moves along a surface of varying height, its momentum allows it to ascend slopes, trading kinetic energy for potential energy.

Hamiltonian dynamics exhibit properties such as time reversibility, Hamiltonian conservation, and volume preservation. Time reversibility ensures that the mapping $T_\epsilon$ from state $(q_t, p_t)$ to state $(q_{t+\epsilon}, p_{t+\epsilon})$ has an inverse $T_{-\epsilon}$ by reversing the time derivatives. This property maintains the desired

distribution invariant during MCMC updates by ensuring reversibility in Markov chain transitions. Hamiltonian conservation is expressed as:

$$\frac{dH}{dt} = \sum_{i=1}^{d} \left[ \frac{dq_i}{dt} \frac{\partial H}{\partial q_i} + \frac{dp_i}{dt} \frac{\partial H}{\partial p_i} \right] = \sum_{i=1}^{d} \left[ \frac{\partial H}{\partial p_i} \frac{\partial H}{\partial q_i} - \frac{\partial H}{\partial q_i} \frac{\partial H}{\partial p_i} \right] = 0$$

For Metropolis updates derived from Hamiltonian dynamics, this conservation implies an acceptance probability of one. The volume conservation property indicates that the volume in the $(q, p)$ phase space remains unchanged:

$$\sum_{i=1}^{d} \left[ \frac{\partial}{\partial q_i} \frac{dq_i}{dt} + \frac{\partial}{\partial p_i} \frac{dp_i}{dt} \right] = \sum_{i=1}^{d} \left[ \frac{\partial}{\partial q_i} \frac{\partial H}{\partial p_i} - \frac{\partial}{\partial p_i} \frac{\partial H}{\partial q_i} \right] = \sum_{i=1}^{d} \left[ \frac{\partial^2 H}{\partial q_i \partial p_i} - \frac{\partial^2 H}{\partial p_i \partial q_i} \right] = 0$$

This property eliminates the need to compute the determinant of the Jacobian matrix.

In practice, Hamiltonian dynamics are approximated via discretization, often using the leapfrog algorithm. HMC has proven to be a highly effective and versatile method for sampling from complex distributions, particularly in high-dimensional spaces, making it a valuable tool for Bayesian inference and other statistical applications (Neal, 2012b).

## 4 METHOD

Overall procedures of our method are described in Algorithm 1. We maintain a collection of measured sequence-score pairs $\mathcal{G}$, and update parameters $\theta$ of the surrogate model $f$ using $\mathcal{G}$ at each round. $f$ consists of an ensemble of $N$ models with same architecture and distinct parameters. In each round, a sampling algorithm based on HMC is employed to propose $K$ new variants $\mathcal{D}_i$ for each $f_i \in f$, and select top $K$ variants for black-box evaluation based on the upper confidence bound over $\mathcal{D}$.

---

**Algorithm 1** Algorithm Overview

---

    **input:** candidates set $\mathcal{D} \leftarrow \{x_{wt}\}$
    Initialize $\theta_i$ for $f_i \in f$
    Initialize ground-truth protein fitness set: $\mathcal{G} \leftarrow \emptyset$
    $x_{best} = x_{wt}$
    **for** $n = 1$ **to** $N$ **do**
        Query ground truth fitness of $X$ and update $\mathcal{G}$: $\mathcal{G} \leftarrow \mathcal{G} \cup \{(x, fitness_x), x \in \mathcal{D}\}$
        Update $\theta_i$ with $\mathcal{G}$ for $f_i \in f$
        Update $x_{best}$ to be the sequence with highest fitness in $\mathcal{G}$
        **for** $f_i$ **in** $f$ **do**
            $\mathcal{D}_i \leftarrow \{x_{best}\}$
            **while** $|\mathcal{D}_i| < K$ **do**
                $\mathcal{D}_i \leftarrow \mathcal{D}_i \cup \text{HMC}(x_{best}, f_i)$
            **end while**
            $\mathcal{D} \leftarrow \mathcal{D} \cup \mathcal{D}_i$
        **end for**
        $\mathcal{D} \leftarrow \text{Top}_K(\text{UCB}(f, \mathcal{D}))$
    **end for**
    **return** $\mathcal{D}$

---

### 4.1 SAMPLING WITH HAMILTONIAN DYNAMICS

**Hamiltonian equation** The Hamiltonian dynamics defines the energy of a particle $q$ by the sum of its potential energy $U(q)$ and kinetic energy $K(p)$, where $p$ is the momentum variable. Here, the $q$ refers to the continuous representation of a protein sequence, initialized with the one-hot representation and then turns into continuous vectors with components constraint between 0 and 1 during the dynamic process. For a sampled $q$, we take the components as the probabilities of amino acids for each site, thus we can discretize the continuous representation $q$ to the one-hot vector $\bar{q}$ by

selecting the amino acids with highest probabilities. The potential energy $U(q)$ represents our target function with higher value represents lower fitness, we define $U(q)$ as the negative log of probability of $f(x)$ and $K(q)$ as the standard kinetic energy:

$$U(q) = -\log P(f(q)), K(p) = \frac{1}{2m}||p||^2 \tag{1}$$

where $f$ is the surrogate fitness prediction model and the probability $P$ is defined as the sigmoid function. We set the mass $m$ to be 1 for all components.

**Time Step Discretization** The approximation of Hamiltonian dynamics is achieved by time discretization defined by the leapfrog algorithm. For a given time $t$, the leapfrog algorithm generate $q_{t+1}$ and $p_{t+1}$ with a half-step update of $p$ followed by a full-step update of $q$ and $p$, as defined in equation 3 and 4.

$$p_{t+\frac{1}{2}} = p_t - \frac{\epsilon}{2}\nabla U(q_t) \tag{2}$$

$$q_{t+1} = q_t + \epsilon p_{t+\frac{1}{2}} = q_t + \epsilon p_t - \frac{\epsilon^2}{2}\nabla U(q_t) \tag{3}$$

$$p_{t+1} = p_{t+\frac{1}{2}} - \frac{\epsilon}{2}\nabla U(q_{t+1}) = p_t - \frac{\epsilon}{2}\nabla U(q_t) - \frac{\epsilon}{2}\nabla U(q_{t+1}) \tag{4}$$

**Virtual barriers as position constraints** The gradient update in equation 3 might results in a $q_{t+1}$ with components exceed the limit of $0 \sim 1$. To handle this constraints, we repeatedly run step 5 after the leapfrog step of equation 3 until $q_{t+1}$ satify the constraint of $0 \leq q_{t+1}(i) \leq 1$. An intuitive illustration is displayed in Figure 2, this procedure establish two virtual barriers during the movement. Hitting the barrier results in a reversed momentum and new position satisfying dynamics. This "bouncing" movement constrains the location of $q$ within valid zones. A theoretical illustration of this mechanism is provided in Appendix B.

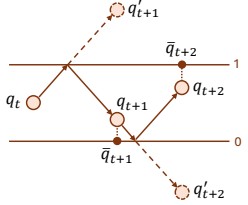

Figure 2: Virtual barrier and position discretization.

$$p_{t+\frac{1}{2}}(i) = -p_{t+\frac{1}{2}}(i), \quad q_{t+1}(i) = \begin{cases} 2 - q_{t+1}(i), & \text{for } q_{t+1}(i) > 1 \\ -q_{t+1}(i), & \text{for } q_{t+1}(i) < 0 \end{cases} \tag{5}$$

**Proposals with Metropolis sampling** Constraints on $q$ provide continuous representation for each amino acid. We take the continuous state as the probability distribution for each site to map back to the one-hot representation. Such approximation inevitably brings error to the discretization. A Metropolis sampling is adopted to reject the samples with large discretization error by discreteized repsentation $\bar{q}_{t+1}$ with probability $\min\{1, \frac{\exp(U(q_t))\exp(K(p_t))}{\exp(U(\bar{q}_{t+1}))\exp(K(p_{t+1}))}\}$, as described in Algorithm 2.

**Candidates election based on Estimated Uncertainty** Gaussian Process (GP) is a preferred choice in Bayesian optimization for its natural property to provide uncertainty estimation based on the covariance. In our method, uncertainty is estimated based on deviations of surrogate model ensembles. Specifically, proposed candidates are filtered based on the upper confidence bound defined by the sum of the average value and standard deviation of model predictions. Different from Ren et al. (2022) which propose samples based on lower confidence bound, samples on upper confidence bound provide refined approximation of high fitness regions with high uncertainty which benefits the Hamiltonian samplers for the exploitation of diverse promising regions.

## 4.2 TWO-STAGE SURROGATE MODEL LEARNING

The surrogate model aims to approximate the ground-truth fitness landscape within a constrained budget of labeled samples. Given that the model inputs are discrete one-hot vectors, it is essential to model the continuous relationships among mutants derived from discrete amino acid variations to facilitate a gradient-based optimization strategy. In this context, protein structure data is utilized as a prior to model these relationships. The fundamental rationale is to leverage the mutual constraints between protein structure and sequence, facilitating the design of protein sequences with similar structures, thereby avoiding being trapped in local optima induced by epistasis effect. Our surrogate model employs an encoder-decoder architecture, and its training proceeds in two distinct stages. The

encoder is utilized in both stages, while two separate decoders are dedicated to learn about protein structure and fitness sequentially.

**Sequence encoder** Inspired by (Hamamsy et al., 2023) that learns structure homology through sequence information, we input a candidate sequence $x$ and apply three pairformer-like shallow attention layers (Abramson et al., 2024) to capture intra- and inter-interactions of amino acid embeddings and produce a latent vector representation, $h(x)$, for each sequence.

**Structure decoder** This first decoder is designed to learn the Root Mean Square Deviation (RMSD) score between a sequence candidate and the wild type across mutation sites. We employ ESMFold (Lin et al., 2023) to predict the structures and compute the conformation perturbations through RMSD scores. During the training phase, the encoder captures the latent representations that reflect the structural distance from input protein to the wild type. Our assumption is that understanding this distribution of structural distances provides prior knowledge of how mutations reflect the structural changes, which benefits the gradient calculation and fitness learning.

**Fitness decoder** In the second stage, the fitness decoder focuses on learning fitness scores, operating with the frozen encoder parameters obtained from the first stage. This separation ensures that the fitness assessment is based on stable, well-defined structural representations derived in the first training stage. Both decoders stack three 1-D convolution layers followed by max pooling and a two multi-layer perceptrons. We did not use complex architectures because these scenarios involve low-resource training, where intricate networks and representations can easily lead to model overfitting.

## 5 EXPERIMENTS

### 5.1 DATASETS, BASELINES AND EVALUATION METRICS

We evaluate our method on two widely recognized combinatorial datasets, GB1 (Wu et al., 2016)and PhoQ (Podgornaia & Laub, 2015) following Wang et al. (2023).Both datasets feature multi-site saturation mutagenesis, each containing approximately $20^4 = 160,000$ samples, specifically covering 149,361 and 140,517 sequences, respectively. Unlike the majority of existing protein fitness datasets, which predominantly include mutants characterized by only one or two mutations, these two datasets avoid relying on inaccurate oracle models for scoring, providing a rigorous testing ground for machine learning-guided directed evolution methodologies.

We compare our method against five baseline approaches: BO refers to Bayesian Optimization with Thompson sampling (BO) based on expected improvement. **CMA-ES**, a classical evolutionary search algorithm, estimates the covariance matrix to adaptively update mutation distributions (Hansen & Ostermeier, 2001). **AdaLead** adopts a straightforward yet robust approach, employing a hill-climbing style to greedily select the top-predicted candidates (Sinai et al., 2020). **PEX** explores the nearby region around wild type by proxmial frontier to trade-off between fitness scores and mutation distances (Ren et al., 2022). **EvoPlay** utilizes a self-play framework within a reinforcement-learning setting, employing Monte Carlo tree search to guide decision-making (Wang et al., 2023). Additionally, we provide the Langevin dynamics version of our methods (HADES-L) for comparison.

We evaluate our methods and baseline models by three metrics: cumulative maximum fitness, average fitness and fitness-conditioned diversity score (fDiv). The maximum and average fitness scores are calculated over $N = 10$ query rounds, with $K = 100$ new queries in each round. The fDiv score is calculated as follows:

$$\text{fDiv}(\mathcal{D}) = \frac{\sum_{(x_i,y_i)\in\mathcal{D}} \sum_{(x_j,y_j)\in\mathcal{D}\setminus(x_i,y_i)} d(x_i,x_j) \cdot (\mathcal{F}(x_i) + \mathcal{F}(x_j))}{2 \cdot |\mathcal{D}| \cdot (|\mathcal{D}| - 1)} \tag{6}$$

where $\mathcal{D}$ denotes the top $K$ samples after $N$ query rounds, and $d(x_i, x_j)$ calculates the edit distance of sequence $x_i$ and $x_j$. The fDiv score follows definition of diversity score from previous studies with a modification to combine the average fitness score of any two different samples, alleviates the dilemma where poor fitness designs reports significantly high diversity score. All results are averaged over 10 runs with different random seeds.

Table 1: Comparing different models for protein engineering task on GB1 and PhoQ datasets. Cumulative maximum fitness, mean fitness and fDiv scores are presented.

| Method | GB1 | | | PhoQ | | |
| --- | --- | --- | --- | --- | --- | --- |
| | max fit. | mean fit. | fDiv | max fit. | mean fit. | fDiv |
| BO | $0.57 \pm 0.15$ | $0.08 \pm 0.01$ | $0.14 \pm 0.02$ | $0.27 \pm 0.05$ | $0.05 \pm 0.01$ | $0.08 \pm 0.01$ |
| CMA-ES | $0.69 \pm 0.16$ | $0.28 \pm 0.07$ | $0.37 \pm 0.08$ | $0.47 \pm 0.23$ | $0.13 \pm 0.03$ | $0.18 \pm 0.03$ |
| AdaLead | $0.84 \pm 0.15$ | $0.49 \pm 0.05$ | $0.64 \pm 0.09$ | $0.62 \pm 0.21$ | $0.19 \pm 0.03$ | $0.26 \pm 0.04$ |
| PEX | $0.83 \pm 0.17$ | $0.51 \pm 0.06$ | $0.68 \pm 0.14$ | $0.46 \pm 0.05$ | $\underline{0.20 \pm 0.01}$ | $\underline{0.30 \pm 0.01}$ |
| EvoPlay | $0.91 \pm 0.09$ | $\underline{0.54 \pm 0.03}$ | $\underline{0.77 \pm 0.04}$ | $0.69 \pm 0.22$ | $0.18 \pm 0.02$ | $0.26 \pm 0.03$ |
| HADES-L | $0.93 \pm 0.14$ | $0.51 \pm 0.06$ | $0.70 \pm 0.12$ | $\underline{0.72 \pm 0.24}$ | $0.20 \pm 0.02$ | $0.28 \pm 0.03$ |
| HADES | $\mathbf{1.00 \pm 0.00}$ | $\mathbf{0.59 \pm 0.02}$ | $\mathbf{0.84 \pm 0.03}$ | $\mathbf{0.80 \pm 0.25}$ | $\mathbf{0.22 \pm 0.02}$ | $\mathbf{0.32 \pm 0.02}$ |

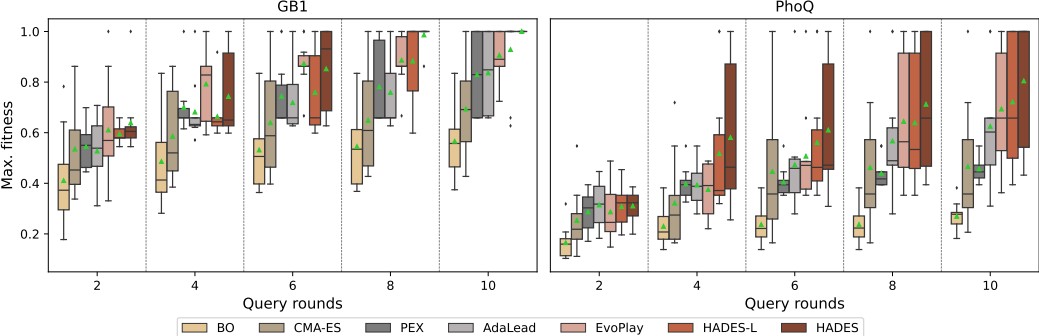

Figure 3: Cumulative maximum fitness scores by experiment rounds on two protein engineering benchmark tasks. Green triangles represent the mean values.

## 5.2 RESULTS AND ANALYSIS

**Overall performance comparison** Table 1 provides a comprehensive evaluation of the performance of our method in comparison to baselines. Bold texts represents best results and the second-best results are marked with underlines. Our method demonstrates superior effectiveness across both tasks on three metrics. Specifically, in the GB1 task, our approach consistently excelled, with all ten experimental runs successfully identifying the optimal protein sequence, providing full score with zero standard deviation. For the more difficult PhoQ task, our method also significantly outperforms baselines on maximum fitness scores. This is indicative of our method's robustness and its effectiveness in navigating more complex problem spaces. Moreover, the best functional diversity (fDiv) scores obtained in both tasks highlight another critical advantage of our method that it not only discover high-fitness variants but also maintain a diverse set of solutions.

To further investigate the optimization curve, we conducted a detailed examination of the optimization efficiency by analyzing the results across various query rounds, as depicted in Figure 3. This investigation allows us to assess the consistency and reliability of our method's performance over successive experimental rounds. Our method provides stable performance across different experiment rounds, persistently outperform baseline methods on both datasets. It is important to note the observation of high variances in the results for the PhoQ task. This variability can be attributed primarily to the inherent sparsity of high-fitness samples within this dataset. Nevertheless, our method's ability to maintain a competitive results highlights its advanced capability to navigate and optimize within intricate and sparse fitness landscapes.

**Ablation experiments** To verify our proposed framework, we conduct a series of ablation experiments, as summarized in Table 2. The baseline model's performance is presented at the top. Subsequent rows detail the impact of different modifications to the framework. We first replace the HMC-based sampling module with proposals based on random mutations. This change led to a marked performance decline in the GB1 task and a moderate decline in the PhoQ task, verifying the

Table 2: Ablation studies for key components.

| Method | GB1 | | | PhoQ | | |
|---|---|---|---|---|---|---|
| | max fit. | mean fit. | fDiv | max | mean | fDiv |
| HADES | **1.00±0.00** | **0.59±0.02** | **0.84±0.03** | **0.80±0.25** | **0.22±0.02** | **0.32±0.02** |
| w/o HD | 0.84±0.15 | 0.52±0.05 | 0.73±0.12 | 0.75±0.25 | 0.21±0.01 | 0.32±0.02 |
| w/o structure | 0.95±0.10 | 0.57±0.04 | 0.84±0.05 | 0.74±0.27 | 0.21±0.02 | 0.31±0.02 |
| w/o ucb | 0.93±0.11 | 0.51±0.04 | 0.75±0.06 | 0.76±0.24 | 0.19±0.02 | 0.30±0.02 |
| w/o vb. | 0.97±0.09 | 0.56±0.04 | 0.79±0.05 | 0.63±0.25 | 0.20±0.02 | 0.30±0.03 |

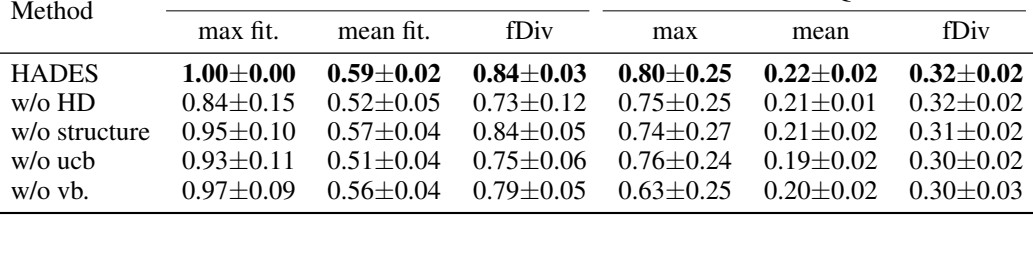

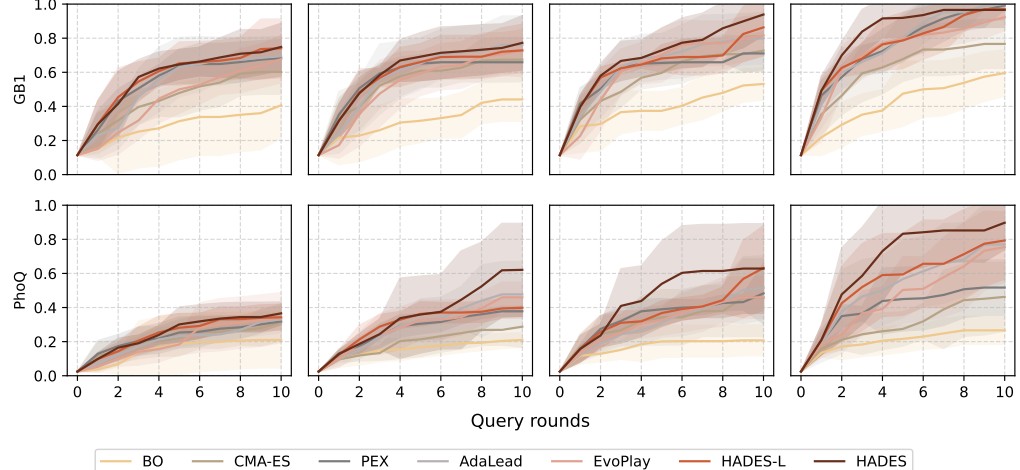

Figure 4: Cumulative maximum fitness scores for K={16, 32, 64, 128} on two protein engineering benchmark tasks. across a suite of protein engineering benchmark tasks. All curves are derived from 10 runs with random network initialization. The shaded regions represent the standard deviation.

effectiveness of our Hamiltonian acquisition strategy. The removal of the structure decoder module is documented in the third row of results, the relatively modest impact on performance could possibly due to generalization error of ESMFold. The fourth row of results involves with omitting the uncertainty estimation process and relying solely on predictions from a single model, this adjustment resulted in uniformly lower scores across all evaluated metrics, confirming the hypothesis that incorporating uncertainty and model ensembles enhances prediction reliability and optimizes performance. The last row of results denotes the modification of removing the virtual barrier mechanism and directly applying discretization. This also led to a reduction in the cumulative maximum fitness score, particularly evident in the PhoQ task's steep landscape. The results demonstrate that the virtual barrier effectively mitigates discretization errors associated with large gradient updates, confirming its utility in enhancing model robustness.

**Effect of query size and query round** In practical protein engineering scenarios, the costs associated with wet-lab evaluations can vary significantly depending on the optimization targets. A robust algorithm, therefore, must maintain stable performance across a range of different evaluation costs. To explore how varying the oracle query size affects algorithm performance, we conducted experiments with varying query sizes $K = \{16, 32, 64, 128\}$ under $N = 10$ rounds of experiments, as shown in Figure 4. Our results indicate that all evaluated methods show improvement as the query size increases. Notably, the performance gap between our method and other methods widens with increasing query size. This observation highlights our method's superior scaling effect, which is particularly pronounced at larger query sizes. The increased performance gap observed at larger query sizes indicates that our method not only adapts well to increased information availability but also leverages this information more effectively than baselines.

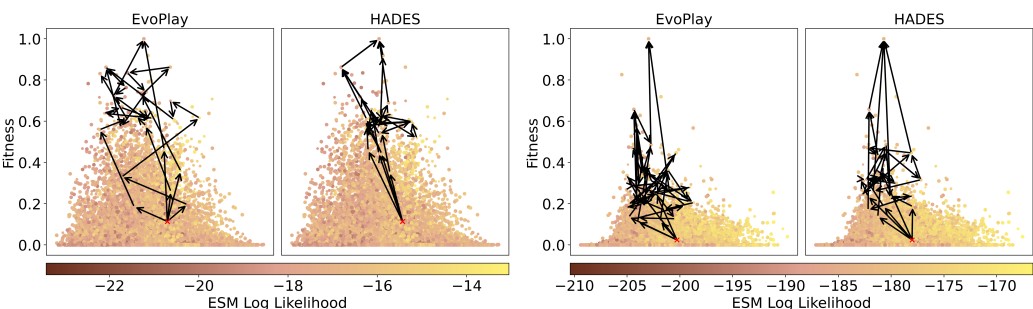

Figure 5: Optimization trajectories of EvoPlay and HADES.

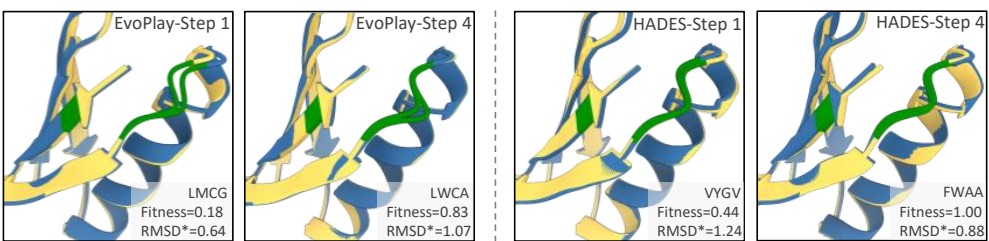

Figure 6: High-fitness structures sampled from EvoPlay and HADES on GB1 task. The RMSD* score (the lower the better) calculates the root mean square deviation over the 4 mutation sites.

**Analysis of sampling trajectory** To provide an intuitive understanding of model behaviours, we visualized the optimization trajectories of EvoPlay and our method, as shown in Figure 5. For this visualization, we utilized the ESM <CLS> token representations of the sampled proteins, projecting these onto a one-dimensional plane via PCA for the horizontal coordinates, with the vertical coordinates representing fitness values. The colors of the dots are determined by the ESM log likelihood, which aggregates the ESM log probabilities of the amino acids. For clarity in the visualization, we plotted the trajectories of the first five runs for each method and task. A trajectory is formed with arrows linking proteins that achieved the maximum fitness scores in each query round. Each trajectory begins at the wild-type, denoted by the red cross, and concludes at the local maximum. For the GB1 task, the trajectories of our method are shorter and more compact, demonstrating enhanced efficiency and robustness in discovering high-fitness variants. On PhoQ task, this comparison is less obvious, however, we observe that our method notably optimizes from the second-best protein directly towards the highest-fitness protein in both tasks, which is not observed with EvoPlay.

**Analysis of structures** To further investigate the effectiveness of our structure learning module. We visualize several wild-type aligned structures with maxmimum fitness sampled at different steps of EvoPlay and our method on GB1 task, as displayed in Figure 6. Proteins colored yellow denotes the wild-type, and proteins colored blue are the model designs. Green regions represents the four mutation sites. From the structure variances in mutation sites, we observe that samples from our method exhibit minor structural changes in mutation region with improved fitness, proving the effectiveness of structure-informed surrogate module.

## 6 CONCLUSION AND FUTURE WORK

We propose a protein engineering framework that combines an acquisition function based on Hamiltonian dynamics and a structure-aware surrogate model for protein directed evolution. In-silico experiments verify that our method efficiently discovers diverse protein variants with high fitness levels, achieving improved efficiency with limited wet-lab costs and experiment rounds. Future efforts will focus on enhancing the modeling of protein structures to improve generalization ability and designing proteins that optimize multiple objectives. Our code will be publicly available upon acceptance.

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

# A HMC-BASED SAMPLING ALGORITHM

Algorithm 2 illustrate our HMC-based sampling algorithm. Note that the Langevin version of our method ignores the update of $p_{t+1}$ in line 5 and reset $p_{t+1}$ as $p_{t+1} \in \mathcal{N}(0, \boldsymbol{I})$.

---
**Algorithm 2** HMC$(q, f)$

---
1: **Input:** seed sequence $q_0$, surrogate fitness scorer $f$
2: Initialize $p_0 \in \mathcal{N}(0, \boldsymbol{I})$
3: $\mathcal{D} \leftarrow \emptyset$
4: **for** $i = 1$ **to** $L$ **do**
5:     Update $p_{t+1}$ and $q_{t+1}$ with formula 3 and 4
6:     Discretize $\bar{q}_{t+1} \leftarrow$ one-hot$(\text{argmax}(q_{t+1}))$
7:     Accept $\bar{q}_{t+1}$ with probability: $\min\{1, \frac{\exp(U(q_t))\exp(K(p_t))}{\exp(U(\bar{q}_{t+1}))\exp(K(p_{t+1}))}\}$
8:     $\mathcal{D} \leftarrow \mathcal{D} \cup \bar{q}_{t+1}$
9: **end for**
10: **return** $\mathcal{D}$

---

# B THEORETICAL ILLUSTRATION OF VIRTUAL BARRIER

For constraint on $p_i \leq 1$, We can split the potential function into two terms: $U(q) = U_*(q) + V_b(q_i)$, where $U_*(q)$ is the potential energy ignoring constraint, and

$$V_b(q_i) = \begin{cases} 0, & \text{for } q_i <= 1 \\ (q_i - 1)b^b, & \text{for } q_i > 1 \end{cases}, \quad \lim_{b \to \infty} V_b(q_i) = \begin{cases} 0, & \text{for } q_i <= 1 \\ \infty, & \text{for } q_i > 1 \end{cases}$$

For constraint that $q_i \geq 0$:

$$V_b(q_i) = \begin{cases} 0, & \text{for } q_i \geq 0 \\ (-q_i)b^b, & \text{for } q_i < 0 \end{cases}, \quad \lim_{b \to \infty} V_b(q_i) = \begin{cases} 0, & \text{for } q_i \geq 0 \\ \infty, & \text{for } q_i < 0 \end{cases}$$

We can thus define Hamiltonian dynamics as:

$$H(q, p) = U_*(q) + V_b(q_i) + K(p)$$

Taking $b \to \infty$, $H(q, p)$ falls back to the original form of $U(q) + K(p)$ for $0 \leq q_i \leq 1$. For $q_i \leq 0$ or $q_i \geq 1$, $H(q, p)$ is dominated by $V_b(q_i) \to \infty$, the $V_b$ term can be seen as a steep hill to climb up until $K(p) = 0$ and $V_b(q_i)$ reaches the value of initial $K(p)$, then the values of $q_i$ falls down to be $0 \leq q_i \leq 1$ and $p_i$ becomes the negative value of original $p_i$, leaving the barrier. Such processes preserve the Hamiltonian properties, providing theoretical high acceptance rate with small discretization error corrected by reject sampling.

# C ADDITIONAL RESULTS

## C.1 RESULTS OF DIFFERENT METHOD USING SAME CNN SURROGATE

We change the surrogate models of PEX, EvoPlay and HADES to use same CNN model for a more fair comparison. Results from table 3 shows that HADES with CNN surrogate still outperforms baselines, proving that our sampler module is capable of proposing promising samples both robustly and efficiently.

## C.2 RESULTS OF UNSUPERVISED METHOD AND GROUND-TRUTH

In accordance with Notin et al. (2023), we evaluate the zero-shot performance of ESM2. The results are assessed based on the top $K = 100$ sequences. As reference, we also include the ground-truth scores which is obtained by directly selecting top $K = 100$ sequences from labeled data. Results

Table 3: Results of different methods on GB1 and PhoQ datasets using same CNN surrogate model. Cumulative maximum fitness, mean fitness and fDiv scores are presented.

| Method | GB1 | | | PhoQ | | |
|---|---|---|---|---|---|---|
| | max fit. | mean fit. | fDiv | max fit. | mean fit. | fDiv |
| BO (CNN) | $0.57 \pm 0.15$ | $0.08 \pm 0.01$ | $0.14 \pm 0.02$ | $0.27 \pm 0.05$ | $0.05 \pm 0.01$ | $0.08 \pm 0.01$ |
| CMA-ES (CNN) | $0.69 \pm 0.16$ | $0.28 \pm 0.07$ | $0.37 \pm 0.08$ | $0.47 \pm 0.23$ | $0.13 \pm 0.03$ | $0.18 \pm 0.03$ |
| AdaLead (CNN) | $0.84 \pm 0.15$ | $0.49 \pm 0.05$ | $0.64 \pm 0.09$ | $0.62 \pm 0.21$ | $0.19 \pm 0.03$ | $0.26 \pm 0.04$ |
| PEX (CNN) | $0.86 \pm 0.16$ | $0.51 \pm 0.05$ | $0.68 \pm 0.11$ | $0.43 \pm 0.03$ | $0.20 \pm 0.01$ | $0.30 \pm 0.01$ |
| EvoPlay (CNN) | $0.85 \pm 0.09$ | $0.48 \pm 0.06$ | $0.67 \pm 0.08$ | $0.58 \pm 0.25$ | $0.15 \pm 0.03$ | $0.22 \pm 0.04$ |
| HADES (CNN) | $0.91 \pm 0.14$ | $0.52 \pm 0.06$ | $0.73 \pm 0.11$ | $0.72 \pm 0.23$ | $0.21 \pm 0.01$ | $0.32 \pm 0.02$ |
| HADES | $\mathbf{1.00 \pm 0.00}$ | $\mathbf{0.59 \pm 0.02}$ | $\mathbf{0.84 \pm 0.03}$ | $\mathbf{0.80 \pm 0.25}$ | $\mathbf{0.22 \pm 0.02}$ | $\mathbf{0.32 \pm 0.02}$ |

Table 4: Results of zero-shot ESM2, HADES and ground truth.

| Method | GB1 | | | PhoQ | | |
|---|---|---|---|---|---|---|
| | mean fit. | fDiv | max fit. | mean fit. | fDiv | max fit. |
| ESM2 | 0.03 | 0.04 | 1.00 | 0.01 | 0.02 | 0.32 |
| HADES | 0.59 | 0.84 | 1.00 | 0.22 | 0.32 | 0.80 |
| Ground truth | **0.65** | **0.98** | **1.00** | **0.37** | **0.63** | **1.00** |

from table 4 shows that zero-shot methods based on ESM2 exhibit almost no signal. In contrast, our method is more closely aligned with experimental results, highlighting the importance of few-shot learning.

# D  IMPLEMENTATION DETAILS

## D.1  IMPLEMENTATION DETAILS

Our model was developed and executed within the PyTorch framework. For the surrogate modeling training, we use Adam optimizer with 0.001 learning rate and MSE loss for both training stages, and the training stops when the training loss does not decrease for 3 epochs. The batch size is 64 during training. The dimension of hidden vector for all neural network moudules is 256, and the kernel size of CNN is 3. For the structure learning module, the RMSD labels define the root mean square deviation on backbone atoms covering C, $C\alpha$, N after aligning the positions of the four mutation sites.

## D.2  SAMPLING DETAILS

We run 128 parallel samples during sampling, the trajectory length $L$ of Hamiltonian dynamics is 16, and the leapfrog step size $\epsilon$ is 0.1 for both tasks. Empirically, the choices of $L$ and $\epsilon$ affect the sampling results significantly. A common strategy is to take preliminary runs to determine the parameters. Here, for fair comparison, we evaluate for the best choices of $L$ and $\epsilon$ on GB1 task and apply on PhoQ task with the same configuration. Following previous works, the first round of sampling is based on the surrogate learning of $K$ mutants with random mutations from wild-type (Ren et al., 2022; Wang et al., 2023).

## D.3  BASELINES DETAILS

Baseline. We conducted a comprehensive comparison of our method against four advanced baseline methodologies:

**Bayesian Optimization**: A classical approach for sequential design problems, it uses an ensemble of models to estimate uncertainty and construct the acquisition function for exploration (Boender, 1991).

**CMA-ES**: A well-known evolutionary search algorithm, CMA-ES adapts the search strategy of subsequent generations by estimating the covariance matrix to update mutation distributions adaptively (Hansen & Ostermeier, 2001).

**AdaLead**: An advanced model-guided evolution technique, AdaLead performs a hill-climbing search on the learned landscape model at each batch query round and then queries sequences with high predicted fitness. This straightforward implementation of model-directed evolution has shown competitive performance against more elaborate algorithms (Sinai et al., 2020).

**PEX**: This method balances fitness scores and mutation distances by exploring regions near the wild type through a proximal frontier. It uses a regularized objective function to search locally for sequences with a low mutation count from the wild-type sequence (Ren et al., 2022).

**EvoPlay**: EvoPlay combines reinforcement learning techniques such as Monte Carlo Tree Search (MCTS) and a policy-value neural network. It focuses on mutating single-site residues in protein sequences, optimizing their properties through an iterative, strategic process akin to gameplay. (Wang et al., 2023)

For BO, CMA-ES and AdaLead, we run implementations provided by FLEXS (Sinai et al., 2020). The surrogate models of these methods share the same architecture with our fitness/structure decoder. For PEX and EvoPlay, we run the official implementations. The baselines are evaluated on a single NVIDIA RTX 4090 GPU with default hyperparameters. We run our method on a single NVIDIA A40 GPU for its compatibility to run ESMFold.

