# OpenReview forum: "Efficient Protein Optimization via Structure-aware Hamiltonian Dynamics"
_ICLR.cc/2025/Conference — ICLR 2025 Conference Withdrawn Submission_

### Official Review · Reviewer_c3be · 2024-10-31

**Soundness:** 1
**Presentation:** 1
**Contribution:** 2
**Rating:** 3
**Confidence:** 5

**Summary:**

HADES is a protein optimization approach that proposes to use Hamiltonian dynamics to sample from a protein sequence distribution with higher fitness and a structure informed prior. A Gaussian Process is utilized to encode uncertainty and filter proposed candidates with a upper confidence bound. HADES is trained and evaluated on GB1 and PhoQ datasets where 4 residues are mutated. They report better performance than certain baselines though I not sure it can be called state-of-the-art based on the chosen baselines since there are more recent and advanced methods.

**Strengths:**

* Using Hamiltonian dynamics is a novel contribution.
* HADES outperforms all chosen baselines on GB1 and PhoQ.

**Weaknesses:**

* The chosen datasets, GB1 and PhoQ, are toyish since they only require mutating up to 4 residues. This is a small search spaces compared to other protein engineering benchmarks such as AAV and GFP [1] that are commonly used in many works. Even the referenced work [2] evaluates on GFP but this dataset is not used. I understand GB1/PhoQ are desirable since they don't require training oracles but there should still be evaluation of realistic protein engineering tasks such as AAV and GFP on top of GB1 and PhoQ. Since the experiments are toyish, we cannot confidently say if HADES outperforms the baselines.
* Several highly related methods are referenced then not compared to. [3, 4] for instance which does diffusion in sequence space seems the most related. Indeed Furthermore, the benchmark is taken from FLEXS which provides more baselines such as CbAS, DynPPO. These baselines are not included. Also [5] is related as it also does search over a learned latent space but is not included.
* While I appreciate the idea of fDiv, weighting the diversity by fitness, it hides the actual diversity of the sequences. The authors should also include the sequence diversity and similarity to the best sequence metrics [1, 5]. It seems odd to not include metrics that have been in related works.
* The technical details are not clear and I am somewhat confused by the method. I will include my technical questions down below.
* I'm not sure what the benefit of Hamiltonian dynamics is over discrete diffusion. It would be good to benchmark as I mentioned. Line 109 states the novelty of HADES over previous methods is doing structure-informed search but as the ablations show this does not significantly improve the results. Therefore the improvement with the novelty of HADES is unclear.

[1] https://arxiv.org/abs/2307.00494
[2] https://www.nature.com/articles/s42256-023-00691-9
[3] https://arxiv.org/abs/2305.20009
[4] https://arxiv.org/abs/2306.12360
[5] https://arxiv.org/abs/2405.18986

**Questions:**

* Line 195. "Query ground truth fitness of X" what is X? I could not find the definition.
* Eq 2-4. What is epsilon?
* Line 185. "f consists of an ensemble of N models with same architecture and distinct parameters." N was previous introduced as the number of iterations then used as the number of surrogate models. This seems wrong? Furthermore, section 4.2 suggests there are two surrogate models but they definitely have different architectures. Are the ensembles just different seeds?
* Section 4.2. I think there needs to be more clarify on exactly what the surrogate model is. There are two decoders but what is the actual potential energy used?

---

### Official Review · Reviewer_Rjwz · 2024-11-04

**Soundness:** 3
**Presentation:** 3
**Contribution:** 2
**Rating:** 5
**Confidence:** 4

**Summary:**

The authors propose an iterative, ensemble based Bayesian Optimization approach for protein sequence optimization. At each optimization round, a new batch of sequences is proposed. The batch is selected from a proposed set according to an ensemble based UCB criterion. The proposed set is generated by starting with the current best variant together with each surrogate model in the ensemble, and running Hamiltonian dynamics with Metropolis-Hastings acceptance of (discretized) proposals. Each surrogate model in the ensemble is updated based on the same oracle scores received in response to the submitted batch.

**Strengths:**

The paper is clearly written. The encoder-decoder architecture aims to distill structure relatedness into the resulting surrogate fitness scores albeit only through shared latent embedding. Nevertheless, bringing some (latent) structural information into sequence optimization seems like a good idea.

**Weaknesses:**

Lots of space is used to discuss Hamiltonian dynamics though this is not strictly speaking followed. HMC(q,f) randomizes the momentum for each call, performs L updates of all residues, starting with the current seed q, accepting each update & its associated discretization with MH. The random momentum moves the system in random direction though remains guided by the potential energy that is defined as -log(P(f(q))). One would think that it would be advantageous to move in the continuous space (relaxation) several steps prior to discretizing the result. Currently, discretization is done after each move which makes connection to the continuous Hamiltonian dynamics also a bit tenuous. This could/should be studied further.

The shared encoder is first trained to predict structural RMSD relative to the wild type. Does this mean that it has to be trained anew for each starting wild type sequence? Also, using aggregate RMSD scores seems a bit strange since RMSD can vary widely in respond to unrelated structural changes (e.g., if ESMFold places a flexible portion in a slightly different position). Not surprisingly from this perspective, the structure decoding guidance didn't seem to help much.

**Questions:**

Which oracle is used to evaluate candidate suggestions?? Despite the larger sequence datasets, a trained oracle is required and should be explained (e.g., in terms of its architectural relation to the surrogate models used in the ensemble).

The authors cite, e.g., Kirjner et al as one of the more recent protein sequence optimization methods but provide no comparison to it. Why? Most of the comparisons are either old or outside the ML literature.

---

### Official Review · Reviewer_HzTQ · 2024-11-04

**Soundness:** 3
**Presentation:** 2
**Contribution:** 2
**Rating:** 5
**Confidence:** 4

**Summary:**

The authors studied Bayesian optimization of protein fitness using a proxy comprised of a sequence encoder, a fitness and a structure decoder, and using a version of Hamiltonian Monte Carlo as a sampling algorithm. The authors showed that this set up can efficiently optimize two tasks, GB1 and PhoQ, efficiently in comparison to some algorithms such as EvoPlay and AdaLead. The authors then showed some ablation studies and optimization trajectory analysis.

**Strengths:**

*Originality*: While HMC is very well-studied in general, and a variety of sequence encoder/decoders have been used as proxies for Bayesian optimization in protein sequences, this combination is (to my knowledge) new. The intuition to using a structure decoder is physically sound and worth exploring (notwithstanding some limitation described below).

*Quality* & *Clarity*: The presentation of the paper is very clear and easy to follow. The analysis is reasonably thorough and well-motivated. The ablation and scaling studies are well appreciated.

*Significance*: The results indeed outperform certain traditional baselines, and this is a reasonable contribution of HMC in protein settings (especially as ablation does show HD helps)

**Weaknesses:**

1. One of the main issues with this paper is the lack of recent baselines and a limited range of tasks with varying difficulties. While HADES appears to be more efficient, it only _marginally_ outperforms the existing baseline methods. Table 3, for example, shows that much of the performance gain (e.g., compared to PEX) can be attributed to an improved surrogate model rather than HMC itself. Moreover, recent literature has demonstrated significant improvements over these baselines. For instance, arXiv:2307.00494 achieves substantially higher fitness than AdaLead/PEX, particularly on more challenging tasks. I would be significantly more convinced if there are either more recent baselines or more difficult tasks.
2. The structure decoder is a focal point in the paper; however, ablation studies indicate essentially _no_ difference in performance without the structure encoder. This aligns with literature findings that ESMFold (and related models) struggle to capture mutational structure differences, particularly for large structural variations. This issue makes the title, introduction, and structural analysis somewhat misleading. Additionally, in practical wet-lab Bayesian optimization cycles, obtaining experimental protein structures will be challenging, casting doubt on the utility of this approach. It might be more beneficial to integrate ESM embeddings instead.
3. The novelty of the paper is limited. While this is admittedly a subjective metric, the paper is primarily an engineering-focused work. Given that each component has been previously studied, strong empirical results are crucial to support the contributions.
4. Related to the novelty concerns, I have some reservations about the paper’s presentation. While the paper is clear, it puts a significant amount of explanation (around 2 pages) on HMC which is not a novel contribution of the paper.

Small issues:
1. There's a missing y-axis in Fig. 3 PhoQ.
2. None of the plots are colorblind unfriendly (especially Fig. 4)

**Questions:**

See above section on 'weakness'

Some small questions:
1. Can the authors show mean fitness instead of max fitness (at least in appendix)?
2. Additionally, could the authors would expand the discussion on limitations?

---

### Official Review · Reviewer_N8VE · 2024-11-06

**Soundness:** 2
**Presentation:** 1
**Contribution:** 2
**Rating:** 3
**Confidence:** 5

**Summary:**

The authors introduce HADES, a protein optimization method based on Hamiltonian Monte Carlo (MCMC), which demonstrates superior performance across in-silico evaluation metrics compared to baseline methods like EvoPlay.

**Strengths:**

The method achieves enhanced performance with fewer sampling steps when tested on two proteins.

**Weaknesses:**

1. The benchmarking is not comprehensive. Experiments are limited to only two proteins (GB1 and PhoQ). More extensive testing is needed to demonstrate the efficacy of the proposed method.
2. The Method section is poorly organized and lacks key details. For example:

   a. There are no details on Bayesian optimization, although it's mentioned in Figure 1 and the Introduction.

   b. The details on the sequence encoder are not described. Do the input features include relative positional encoding? What are the hyperparameter settings for each Pairformer block, such as the number of heads in the attention layer?

   c. On Page 6, line 275, the term "intra-interactions" of amino acid embeddings is unclear. The "inter-interactions" makes sense, but "intra-interaction" seems awkward in this context.

   d. On Page 6, line 275, the paper claims that the sequence encoder produces a latent vector. However, the Pairformer outputs single representations for each amino acid and pair representations for amino acid pairs. How are these representations combined into a single latent vector?

    e. On Page 4, line 206, what does UCB stand for? This acronym is not defined anywhere in the paper.

    f. While the overall architecture (Figure 1) is described in the Introduction, it is not adequately explained in the Method section.

3. Key theoretical analysis is missing. The paper states that Metropolis sampling is applied to correct errors from discretization (Page 5, line 250). How much additional computational overhead does this introduce?

4. Important related work on MCMC in discrete spaces is absent. Several methods have applied Langevin MCMC to sample discrete sequences[1,2], which should be discussed.

In summary, given its preliminary results and issues with clarity and organization, this paper appears to be more suitable for presentation at a workshop. It may not yet meet the standards expected for a conference publication.

References
1. Zhang et al. A Langevin-like Sampler for Discrete Distributions. ICML 2022. https://proceedings.mlr.press/v162/zhang22t.html

2. Sun et al. Discrete Langevin Samplers via Wasserstein Gradient Flow. ICML 2023. https://proceedings.mlr.press/v206/sun23f.html

**Questions:**

Please addresss the points in the Weakness section.

---

### Note · Authors · 2024-11-18

I have read and agree with the venue's withdrawal policy on behalf of myself and my co-authors.